# Performance Analysis of Software-Defined Networks to Mitigate Private VLAN Attacks

**DOI:** 10.3390/s23041747

**Published:** 2023-02-04

**Authors:** David Álvarez, Pelayo Nuño, Carlos T. González, Francisco G. Bulnes, Juan C. Granda, Dan García-Carrillo

**Affiliations:** Department of Computing, University of Oviedo, Campus de Viesques, 33204 Gijón, Asturias, Spain

**Keywords:** software-defined networks (SDNs), private VLAN (PVLAN), security, segmentation

## Abstract

The defence-in-depth (DiD) methodology is a defensive approach usually performed by network administrators to implement secure networks by layering and segmenting them. Typically, segmentation is implemented in the second layer using the standard virtual local area networks (VLANs) or private virtual local area networks (PVLANs). Although defence in depth is usually manageable in small networks, it is not easily scalable to larger environments. Software-defined networks (SDNs) are emerging technologies that can be very helpful when performing network segmentation in such environments. In this work, a corporate networking scenario using PVLANs is emulated in order to carry out a comparative performance analysis on defensive strategies regarding CPU and memory usage, communications delay, packet loss, and power consumption. To do so, a well-known PVLAN attack is executed using simulated attackers located within the corporate network. Then, two mitigation strategies are analysed and compared using the traditional approach involving access control lists (ACLs) and SDNs. The results show the operation of the two mitigation strategies under different network scenarios and demonstrate the better performance of the SDN approach in oversubscribed network designs.

## 1. Introduction

The number of services and interconnected devices has risen significantly in recent years due to digital transformation and the advent of new technologies such as cloud computing and the Internet of Things (IoT). Therefore, network administrators must cope with several security threats that can be unintentionally introduced within the corporate network, leading to vulnerable exposed systems and services. The tasks performed by network administrators tend to be complex and usually prone to errors, thus compromising network security. In fact, some attack techniques leveraging non-existent or bad configurations in layers 2 and 3 of the Open Systems Interconnection (OSI) model have been traditionally used by cyber adversaries [1,2].

The defence-in-depth (DiD) methodology is a defensive approach traditionally implemented by network administrators to design and build secure networks by layering and segmenting them [3]. Network segmentation relies on organising corporate resources in zones with similar security requirements [4,5]. Then, a firewall enforcing restrictive policies is deployed between zones. Typically, segmentation is implemented in layer 2 using standard virtual local area networks (VLANs) or private virtual local area networks(PVLANs). Although DiD may well suit the needs of a conventional corporate network with reduced size, it is not easily scalable to larger environments since it is difficult to manage, requires extensive skill sets, and can be extremely costly [6]. Therefore, some authors state that this defensive model composed of several layers representing different security tactics is outdated [7].

Software-defined networks (SDNs) are emerging technologies with a wide variety of innovative application areas such as network virtualisation (NV) in cloud environments [8], time-sensitive networking (TSN) [9], vehicular ad hoc networks (VANETs) [10], and distributed deep learning (DDL) [11]. Within the area of security, SDNs may help reduce the management complexity of DiD in large corporate networks, for example, when performing network segmentation [12,13,14,15]. SDNs introduce a new networking paradigm to manage data and control planes separately. SDNs are based in a central node, called the SDN controller, which has full knowledge of the data plane. The data plane is composed of network devices, which are in charge of commuting or routing each packet properly. In addition, the SDN controller is responsible for the control plane, so it manages the deployment, organisation, and behaviour of the elements of the data plane. To do so, the SDN controller performs software-based decisions, which implies expanding the application of dynamic data-flow tables and rules to the rest of the network devices. As the SDN controller is a programmable entity, advanced security control and deceptive strategies can be implemented to increase network security.

Since the rise of SDNs, numerous research works have been published comparing the conventional networking approach with the SDN-based approach. The results of these studies show that SDNs have lower delay and jitter, as well as a higher throughput than conventional networks [16]. In addition, routing convergence time with large topologies is shorter in SDNs than in conventional networks, indicating that SDNs converge faster [17]. Following this research topic, it is also interesting to study how the implementation of security measures may benefit from the SDN approach in terms of network performance and resource usage.

In this research, a corporate networking scenario using PVLANs is emulated using two mitigation strategies: a conventional approach and an SDN-based approach. PVLANs are secured in the conventional approach by configuring access control lists (ACLs) in the gateway of the corporate network. In contrast, in the SDN-based approach, security rules are configured in the switches of the LAN segment. Then, the network is attacked to carry out a comparative performance analysis on mitigation strategies when facing attacks in both deployments. To that end, a well-known PVLAN attack is executed by emulating an attacker located within the corporate network. Several performance parameters such as CPU and memory usage, communications delay, packet loss, and power consumption are evaluated in network devices.

The rest of the paper is organised as follows: Section 2 briefly describes the working principles and security concerns of PVLANs. Related work is discussed in Section 3. In Section 4, the performance comparative analysis performed between the conventional and SDN-based mitigation strategies is detailed. In Section 5, the results of the experimentation are presented. Finally, Section 6 contains the concluding remarks and outlines future work.

## 2. Private VLAN

Private VLAN (PVLAN) is a technique that can be used to segment and adapt a broadcast domain. PVLANs are used to isolate endpoints at layer 2, so direct communication among endpoints belonging to the same IP network is not possible. PVLANs are configured on switches by assigning ports to one of the following three roles: promiscuous, community, or isolated. Firstly, endpoints connected to a promiscuous port can communicate at layer 2 with endpoints connected to promiscuous, community, and isolated ports. Secondly, endpoints connected to a community port can only communicate at layer 2 with endpoints connected to promiscuous ports and ports belonging to the same community. Thirdly, endpoints connected to an isolated port can only communicate at layer 2 with endpoints connected to promiscuous ports. An example of a configuration of a PVLAN is shown in Figure 1, where there are two promiscuous ports, two different communities, and three isolated ports. Note that trunk links allowing for extending the scope of the PVLAN across several switches are omitted for the sake of clarity.

As observed in the figure, *P1* and *P2* are promiscuous ports. They can communicate with all the community and isolated ports and also with each other. The network gateway is usually connected to a promiscuous port. *C1* and *C2* are two different communities composed of six and four ports, respectively. Communications involving ports within the same community, as well as with promiscuous ports, are allowed, but ports in *C1* cannot communicate with those in *C2* and vice versa. Finally, the three isolated ports (*I1*, *I2*, and *I3*) can communicate with *P1* and *P2* exclusively, since communication with another type of port, even between two isolated ports, is not allowed. It should be noted that isolated, community, and promiscuous ports cannot communicate with default ports of the switch, that is, ports with no PVLAN configuration (the remaining nine ports in the figure).

A simplified variant of PVLAN is known as PVLAN edge. PVLAN edge uses only two types of ports: protected and unprotected. The ports of the switch are configured as unprotected by default. In contrast, protected ports must be explicitly configured. Communication at layer 2 between endpoints connected to protected ports is not allowed. An example of a PVLAN edge is shown in Figure 2. As can be seen, there are eight endpoints connected to protected ports, and the gateway is connected to an unprotected port. According to this configuration, network users can only communicate with the gateway.

### Private VLAN Attack

The main threat related to both PVLAN and PVLAN edge is private VLAN attacks, which may affect both the physical and virtual layer 2 architectures [18]. They allow for sending unidirectional traffic while avoiding the restrictions imposed by PVLANs. For example, a device connected to an isolated port may reach a device connected to a community port. Similarly, a device connected to a community port may send traffic to a device connected to a port belonging to a different community. An example of a PVLAN attack that exploits this vulnerability is shown in Figure 3.

In this example, a user connected to the port labelled as *Isolated A* is an attacker who aims to send malicious unidirectional traffic to *User B*, who is a member of the $C_{1}$ community. As illustrated in Figure 3a, the attacker crafts packets whose destination IP address is the IP address of *User B*, while the destination MAC address is the MAC address of the network gateway. This initial step is legitimate since communications between isolated ports and promiscuous ports are allowed in PVLANs. When the gateway receives the crafted packets, it routes them to *User B*, as depicted in Figure 3b. Since the gateway knows the path towards the destination (*User B*) because both are located in the same network, the gateway forwards the crafted packets after changing the destination MAC address for the MAC address of *User B*. This final step is possible since communications from a promiscuous port to a community port are also allowed in PVLANs. Thus, the attacker can evade the restrictions imposed by PVLANs between different types of ports by relying on a promiscuous port.

The traditional solution to mitigate the PVLAN attack in conventional networks is to configure access control lists (ACLs) in the network gateway [19]. Specifically, an access control rule should be implemented where all communications having both source and destination IP addresses within the LAN are denied, with the exception of the case when the destination IP address is that of the network gateway. This ACL must be applied to inbound traffic on the interface that the network gateway uses in the LAN. This approach is also applicable if a firewall is located between network levels, although the ACL in this case would be replaced by a firewall rule. Moreover, an intrusion prevention system (IPS) can be placed within the network to stop the PVLAN attack, but a networking rule similar to the ACL is needed to discard the traffic. In both cases, the operating principle is the same as for the traditional solution.

## 3. Related Work

Several research works have been published aiming to improve the deployment of conventional network segmentation solutions. In [20], the authors present *the simple-set-based* algorithm, a mathematical solution that creates an efficient VLAN network structure by analysing traffic flows and defining which VLAN is the most suitable for each node of the network. Another VLAN segmentation approach is proposed in [21] to adequately separate voice and data packets in Voice-over-IP (VoIP) networks to improve network security.

Although the use of VLANs is a widespread solution in conventional corporate networks, it poses some disadvantages such as a more complex network addressing design and the potential presence of vulnerable VLAN-related protocols such as Dynamic Trunk Protocol (DTP), VLAN Trunking Protocol (VTP), or Generic VLAN Registration Protocol (GVRP). To overcome some of these drawbacks, PVLANs have been used as an alternative in order to secure and optimise layer 2 communications both in industrial [22] and corporate networks [23]. In addition, PVLANs have been proposed as an effective network isolation technique to protect IoT environments [24]. Nowadays, PVLANs are mainly used by Internet Service Providers (ISPs) to avoid communications between different customers within the same LAN segment, in business parks where companies share the same public IP address space, and in shared hosting sites [25].

Leaving the security component aside, PVLANs together with SDNs have also been used in cloud computing environments to simulate link aggregation to enhance performance. For example, PVLAN promiscuous ports are combined as a single logical port to optimise load balancing and path selection [26]. Although the SDN paradigm cannot be directly extrapolated to conventional corporate networks, its goals and some of its fundamentals can. In fact, as an SDN can be programmed to detect anomalous traffic and subsequently modify the network behaviour, there is extensive related work focused on the detection of potentially dangerous traffic and the mitigation of the risks associated with it.

A solution to detect and mitigate Address Resolution Protocol (ARP) spoofing and ARP poisoning attacks are described in [27]. An SDN security module designed to stop ARP Spoofing attacks within corporate networks with minimal intervention from network administrators is presented in [28]. A similar solution tailored to cloud-fog-edge platforms can be found in [29]. In addition, a solution to mitigate Dynamic Host Configuration Protocol (DHCP) starvation and DHCP spoofing attacks is described in [30]. Finally, there are other interesting works focused on protecting the network from other less common attacks such as flooding attacks through ARP storming [31] and Neighbor Discovery Protocol (NDP) spoofing [32].

Recently, a large number of research works have been published focused on performing comparative analyses on SDNs and conventional networks. These works evaluate several aspects such as routing protocols [16,17], control plane security [33], and delay [34,35], among others. Regarding PVLANs, a comparison between an SDN-based model and a conventional network is presented in [36], where the deployment of VLAN and PVLAN filtering and segmentation using both approaches is analysed. The authors state that the conventional approach can be difficult to configure in large network environments, which implies a higher deployment cost. They also conclude that an SDN solution is easier to implement, as the intelligence is centralised in the SDN controller. However, they do not carry out a performance evaluation of both approaches.

In this work, a corporate networking scenario using PVLANs is emulated, and different mitigation strategies are evaluated in order to carry out a performance comparative analysis on the use of resources of both strategies. To the best of the authors’ knowledge, this is the first study that compares the performance of a conventional solution and an SDN-based solution to secure a corporate network where PVLANs are deployed as a network segmentation mechanism.

## 4. Comparative Performance Analysis

In this section, several aspects regarding the comparative performance analysis are explained: the mitigation strategies that were compared, the testbed used during the experimentation, and the test procedure.

### 4.1. Mitigation Strategies

Two mitigation strategies to face the PVLAN attack were evaluated in this analysis. In order to describe them, 192.168.1.0/24 is used as the network address of a LAN segment and 192.168.1.1 is the IP address of the network gateway.

Figure 4a shows the rules that compose an ACL that can be configured in the gateway in a conventional solution. They block all communications that have source and destination IP addresses within the same network, except those where the destination IP address matches that of the gateway. Note that this ACL must be applied to process inbound traffic in every interface (both hardware and software) that is acting as the gateway, or aggregation point, of a LAN segment.

A solution based on the use of SDNs was also evaluated. The OpenFlow protocol version 1.3 was used in these tests in combination with the *Ryu* [37] SDN controller. The OpenFlow communications between the controller and the switches were performed through an out-of-band (OOB) management network. In this approach, the controller uses the OpenFlow protocol to proactively install the necessary rules on the flow table of the switches to discard all traffic that matches the pattern of a PVLAN attack. The controller only manages the access level switches because, in a corporate network, the access level is the first line of defence against internal attacks, such as the PVLAN attack. Thus, this prevents traffic from reaching other areas of the network, resulting in better network performance as malicious traffic is dropped from as close to the attacker as possible [38].

Figure 4b shows the rules that compose the flow table of the switches. The communications with source and destination IP addresses within the same network are discarded, except those where both the destination IP address and the destination MAC address match those of the network gateway. To do so, two flows are installed on the flow table of the access level switches. First, the flow with priority value *4* permits legitimate traffic from the LAN segment to the network gateway. If this flow is matched, the packet is transmitted with no further processing since it is the highest priority flow. Second, the flow with priority value *3* blocks traffic with source and destination IP addresses within the LAN segment and the destination MAC of the network gateway, since this corresponds with the PVLAN attack pattern.

### 4.2. Testbed

The comparative analysis carried out in this work was performed in a testbed that emulated a small corporate network with a collapsed core network design. Two network scenarios were evaluated, as shown in Figure 5. Both scenarios are examples of oversubscribed network designs since the switches have more downlink bandwidth available than uplink bandwidth [39]. The first scenario comprises three different LAN segments with each switch (*S1*, *S2*, and *S3*) connected to the collapsed core level. The second scenario represents a single LAN segment composed of two switches at the access level (*S1* and *S3*) and another switch (*S2*) acting as an aggregation switch connected to the collapsed core level. The second scenario was proposed due to the large difference in computational resources between the available router and switches. Thus, this will ensure that the behaviour observed during the experimentation does not depend on the network devices used.

During the tests, a Cisco 1841 router with two 100 Mbps interfaces and an HWIC-4ESW module with 4 Ethernet ports was used as a gateway in the collapsed core level. In addition, three Netgear ProSAFE M4300-28G switches with twenty-four 1 Gbps interfaces and four 10 Gbps interfaces were deployed at the access/aggregation level.

The corporate network emulated in the two scenarios also included PVLAN segmentation, as illustrated in Figure 5. Specifically, all the switches located at the access level were configured with two communities (blue and yellow) with six ports reserved for each community and six isolated ports (red). In the first scenario, a promiscuous port was configured on the switches in the switch-to-gateway links. In addition, in the second scenario, where it was necessary to extend the PVLAN segmentation, trunk links were configured between switches. Note that in this scenario the ACLs were configured in the aggregation switch and applied inbound in its trunk links.

### 4.3. Experimentation

To analyse the performance of the two mitigation strategies, a variable bandwidth of legitimate and malicious traffic between endpoints and their gateway was injected into the network. This traffic used UDP as the transport protocol and had to be processed by the ACL configured in the gateway or on the flow tables of the switches, depending on the strategy for detecting PVLAN attacks. UDP packets did not exceed 1500 bytes to adjust to the standard Ethernet frame size. The legitimate traffic used a fixed bandwidth of one percent of the available bandwidth on the uplinks in the switches located at the access level in each scenario. Therefore, the legitimate traffic used 1 Mbps in the first scenario and 10 Mbps in the second scenario and was generated from one member located in each community.

Malicious traffic was generated from endpoints located in isolated ports in both scenarios. In the first scenario, one attacker was emulated in each of the three LAN segments, and the aggregate malicious traffic was progressively increased in a balanced manner by 15 Mbps (5 Mbps per attacker) between tests. In the second scenario, two attackers were emulated in *S1* and two attackers in *S3* switches, and the aggregate malicious traffic was progressively increased in a balanced manner by 400 Mbps (100 Mbps per attacker) between tests.

During the tests, the corporate network supported an aggregated background traffic of 8 Gbps exclusively at the access level in both scenarios. This traffic simulated a multimedia e-learning activity on the corporate network via multicast communications. It should be noted that the multicast traffic involved four members of the two communities, so this traffic was neither received nor sent through isolated ports. In addition, both scenarios were configured so multicast traffic was not sent through the promiscuous ports of the switches, so the network gateway (Figure 5a) or the aggregation switch (Figure 5b) never processed this type of traffic. The *iperf* traffic injector was used [40] to generate the legitimate, and the malicious UDP traffic processed by the mitigation strategies and the multicast background traffic that emulated the underlying e-learning activity.

Finally, several tests were performed in the router and the switches in both scenarios in order to measure several performance parameters such as CPU, memory usage, and power consumption. The round-trip delay time and packet loss were measured in a non-attacker user connected to an isolated port. Tests had a duration of five minutes and were repeated three times. Table 1 summarises the settings used during the experimentation.

## 5. Results

The results of the experimentation in the multiple- and single-LAN scenarios are presented in Section 5.1 and Section 5.2, respectively.

### 5.1. Multiple-LAN Scenario

Figure 6 shows the average CPU usage by the network devices as the aggregated malicious traffic increases. As observed in Figure 6a, the CPU usage grows faster at the core level as the ACL configured in the router processes more traffic. When the aggregate malicious traffic reaches 90 Mbps, CPU usage peaks and stabilises at around 50%. This means that the router begins to discard traffic when exceeding 90 Mbps without processing it. Figure 6a also shows how the CPU usage in the core level is much lower, remaining constantly at around 7%, when the mitigation strategy is based on SDNs, since the malicious traffic is detected and blocked by the switches located at the access level. Figure 6b shows the CPU usage by the switches. As can be seen, the CPU usage slightly increases when the mitigation strategy is based on SDNs, since the switches assume extra workload. These plots demonstrate how the workload moves from one level to another according to the mitigation strategy deployed in the scenario.

Moreover, the average memory consumption by the network devices using both mitigation strategies is shown in Figure 7. As with the CPU usage, the memory consumption in the router located at the core level is higher when the mitigation strategy is based on ACLs than when it is based on SDNs as shown in Figure 7a. In contrast, Figure 7b shows that the memory consumption in the switches located at the access level is slightly higher with the strategy based on SDNs. However, there is no significant difference when comparing mitigation approaches, since the memory consumption is hardly affected by the increment of malicious traffic between tests.

Figure 8 shows how a legitimate user is affected by the implementation of each mitigation strategy. When the strategy is based on ACLs, a PING from a legitimate user to the gateway takes longer as the malicious traffic increases, as can be seen in Figure 8a. The average round-trip time delay stabilises at 105 Mbps of aggregate malicious traffic. This means that the router begins to discard traffic without processing it. In contrast, the round-trip time delay remains constant when the mitigation strategy is based on SDNs. The results show how each approach affects this scenario. In the case of the ACL-based approach, the filtering of malicious traffic is achieved at the router, whereas in the case of the SDN-based approach, the filtering is performed at the switches. Thus, when using SDNs, malicious traffic is prevented from reaching the router, resulting in a lower RTT.

Figure 8b shows the average percentage of packets lost when pinging from a legitimate user to the gateway. As discussed for the CPU usage and the round-trip time delay for the ACL-based approach, it can be seen that the router discards packets without processing them starting from 105 Mbps of aggregate malicious traffic, so it does not discriminate between legitimate and malicious traffic. In contrast, no packet loss occurs with the SDN-based approach. These results demonstrate that, in this scenario, using the mitigation strategy based on configuring ACLs at the core level has a more negative impact on legitimate users than using the mitigation strategy based on SDNs.

Finally, the average power consumption for each mitigation strategy is shown in Figure 9. In the ACL-based approach, at the core level, the router consumes slightly more power because the CPU usage is much more intensive than in the SDN-based approach. On the other hand, since the switches are not operating intensively, there is no significant difference at the access level, regardless of the mitigation strategy used.

### 5.2. Single-LAN Scenario

In the network design used in the previous scenario, there was an imbalance between the computational and network resources of the devices deployed at both levels. Specifically, the router located at the core level had fewer resources than the switches located at the access level. Therefore, in this scenario, a more balanced design is used, where mitigation strategies are applied at the aggregation and access levels using devices with similar characteristics. The objective is to determine whether the same behaviour is observed in this scenario as in the previous one.

The average CPU usage by the network devices as the aggregated malicious traffic increases is shown in Figure 10. The CPU usage in this scenario is always higher at the access level than at the aggregation level. This is due to the exchange of background multicast traffic since the access level switches transmit a greater number of multicast flows than the aggregation level switch.

As can be seen in Figure 10a, there is slightly higher CPU usage at the aggregation level when the mitigation strategy is ACL-based. This is because the switch deployed at that level must process all the malicious traffic. When the mitigation strategy used is SDN-based, the CPU usage decreases at the aggregation level and moves to the access level as shown in Figure 10b. The two switches suffer an increase in CPU usage due to traffic analysis on the flow table. Therefore, these results again demonstrate how the workload moves between levels depending on the mitigation strategy deployed.

Figure 11 shows the average memory consumption by the network devices depending on the mitigation strategy used. As in the previous scenario, memory consumption is hardly affected by the increment of malicious traffic between tests. As observed in the figure, the memory consumption in both levels is slightly higher when the mitigation strategy is based on SDNs. However, the difference between approaches can be considered negligible.

Figure 12 shows the average round-trip delay time of a PING from a legitimate user to the gateway as malicious traffic increases using both mitigation strategies. As can be seen, neither of the mitigation strategies has a better impact on latency, which varies from 2.3 to 2.5 ms, with no significant difference between the ACL-based and the SDN-based approach. In addition, there was no packet loss in either approach during the tests, as this scenario does not come close to overloading the switches.

Finally, the average power consumption for each mitigation strategy is shown in Figure 13. As can be seen, there is no significant difference at either level, regardless of the mitigation strategy used.

#### Worst-Case Scenario: Faulty ACL Configuration

The two scenarios used during the experimentation are examples of oversubscribed networks. In the multiple-LAN scenario, the access switches have more bandwidth available in their downlinks than in their uplinks, while in the single-LAN scenario, the same is true for the aggregation switch. This oversubscribed network design, combined with an ACL configured in outbound mode, may lead to inefficient performance due to a bottleneck. This issue is described and evaluated using the single-LAN scenario since the traffic, both legitimate and malicious, coming from the downlinks of the aggregation switch to the router is significantly higher than in the multiple-LAN scenario.

An outbound ACL analyses traffic after it has been routed/commuted. Packets are processed through the outbound ACL after being sent to the outbound interface. This approach is useful when the same criteria must be applied to the traffic received in multiple interfaces whose destination is the same outbound interface. In the single-LAN scenario, there were two trunk ports configured the in 10 Gbps interfaces of the aggregation switch. In addition, these two ports had inbound ACLs configured. With a higher number of trunk links, an outbound ACL in the promiscuous interface of the aggregation switch would appear to be an intelligent design decision because it would simplify ACL placement. However, since the traffic forwarding is carried out by the switching hardware, and the interface configured as a promiscuous port is limited to 1 Gbps, traffic would be processed by the ACL statements more slowly than with the current inbound ACL configuration. The use of SDNs always enables traffic to be analysed before it is sent to the outbound interface, resulting in proper performance.

Figure 14a shows that a PING from a legitimate user to the gateway takes longer when an outbound ACL is configured in the promiscuous port of the aggregation switch. Although the difference with other approaches is not large, it is significant. However, the number of lost packets shown in Figure 14b is unacceptable. As observed in the figure, as malicious traffic increases, the percentage of lost packets rises, while there is no packet loss in the other approaches. It is particularly interesting to note that this effect does not occur until 1.2 Gbps of aggregate malicious traffic is reached, which is slightly higher than the interface bandwidth and well in excess of the actual link speed of 100 Mbps. Therefore, this faulty configuration could go unnoticed, depending on the network usage.

In this regard, using the SDN-based approach eliminates the possibility of making the wrong design decision described in this section.

## 6. Conclusions

In this paper, two strategies to mitigate the PVLAN attack based on ACLs and SDNs were analysed and compared. Both strategies were tested in scenarios with different two-level network designs and with network devices with different levels of performance. Several parameters such as CPU and memory usage, round-trip time, packet loss, and power consumption were evaluated. The results showed that the proposed mitigation strategy had an impact on these parameters.

The conventional ACL-based approach tended to increase CPU consumption in higher-level devices, while the SDN-based approach did so in lower-level devices, in a more balanced way. Memory usage also showed this behaviour, but the difference between the two approaches could be considered negligible. In terms of network performance from the perspective of users, there were no significant differences between mitigation strategies. However, there was a very large difference in an oversubscribed network scenario if ACLs were placed incorrectly. The effect was a significant increase in the round-trip time, while packet loss quickly rose to 50%. It is also worth noting that both scenarios had a limitation, as no additional measures such as a firewall, an intrusion detection system (IDS), or an intrusion prevention system (IPS) were in place during testing. However, the inclusion of additional systems would not affect the performance analysis carried out in this paper.

Therefore, in an oversubscribed scenario, the SDN approach prevents the network administrator from making mistakes in the configuration of the ACLs, specifically, in which interface to place them, so the SDN approach ensures proper operation and performance. Furthermore, the use of an SDN-based mitigation approach allows for a wider range of deception strategies. Not only does this approach rely on blocking forbidden communications, but malicious packets can also be sent to a honeypot, or even sent back to the attacker by setting the inbound port as the outbound port in the action list regarding the PVLAN attack flow in the flow table. Additional benefits of using the SDN approach are that resource consumption can be balanced at the access level, so bottlenecks can be avoided, and that configuration is simpler and less prone to errors than using an ACL approach.

Future work will focus on comparing the performance of SDNs with conventional network mitigation strategies to protect corporate networks against other common types of attacks. Examples are SDN-based alternatives to implement defensive strategies at the access level switches to protect against flooding or to avoid sending unicast frames to unknown MACs.

Similarly, the development of SDN-based defensive strategies for other network environments, such as IoT networks, will be explored. In particular, research is underway on a system that detects IoT devices that are deployed on different ports than expected in a switched environment, and correcting this configuration. The system makes it possible to modify the configuration of the switch ports if the device is well known, to accommodate it if the device is new, or to exclude it from the network if any anomalous behaviour is detected.

## Figures and Tables

**Figure 1 sensors-23-01747-f001:**
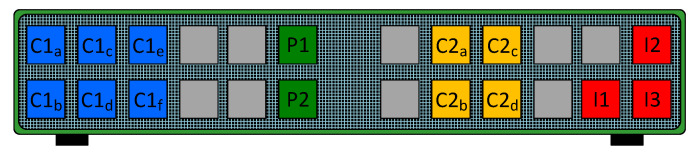
PVLAN configuration on the switch.

**Figure 2 sensors-23-01747-f002:**
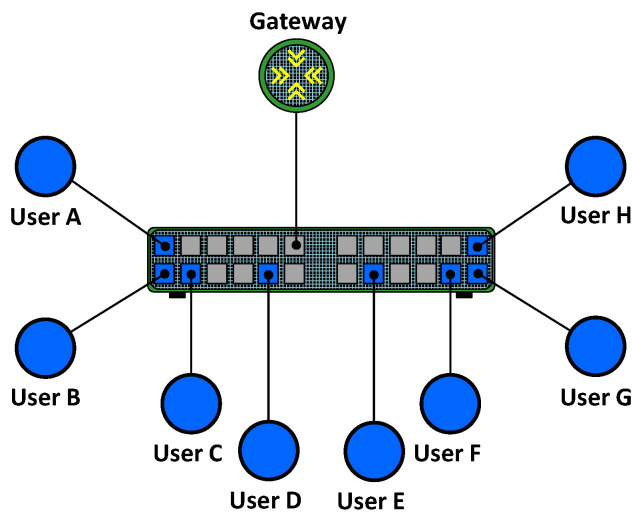
PVLAN edge.

**Figure 3 sensors-23-01747-f003:**
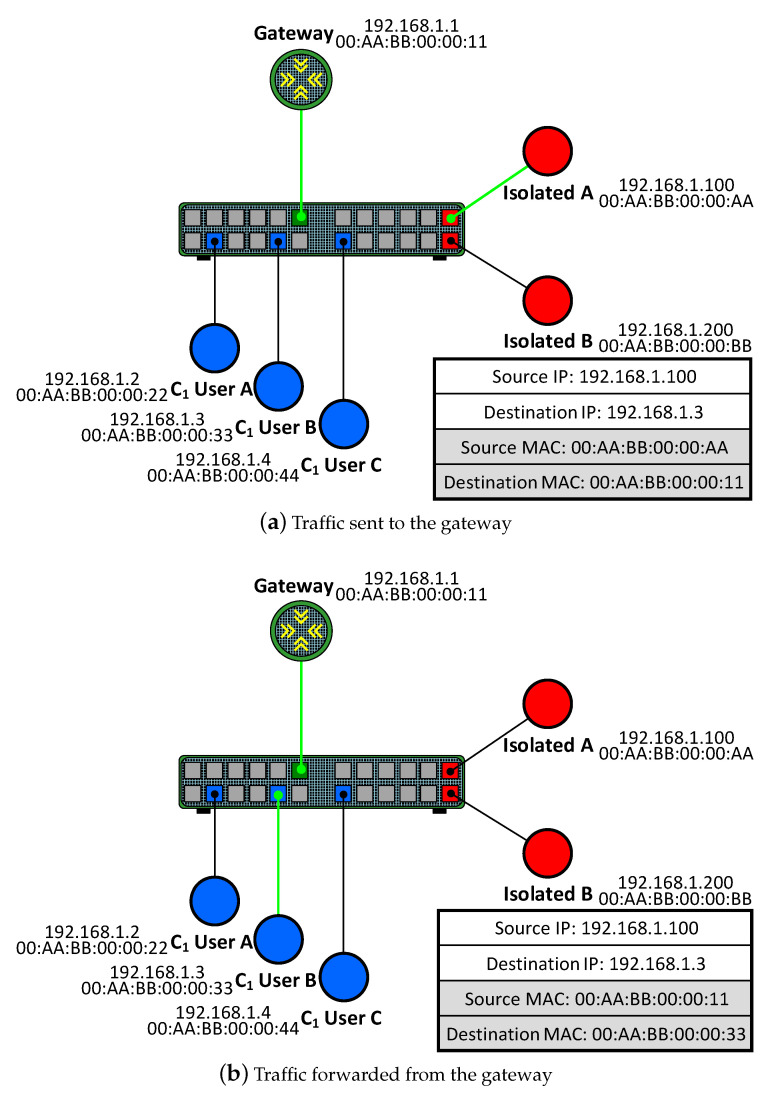
PVLAN attack.

**Figure 4 sensors-23-01747-f004:**
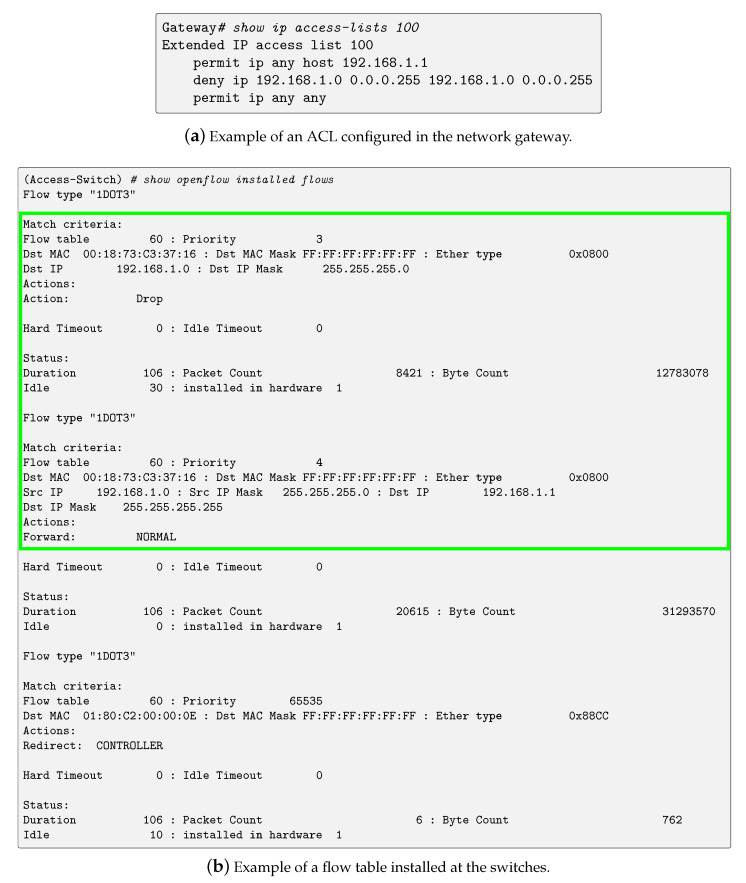
PVLAN attack mitigation strategies.

**Figure 5 sensors-23-01747-f005:**
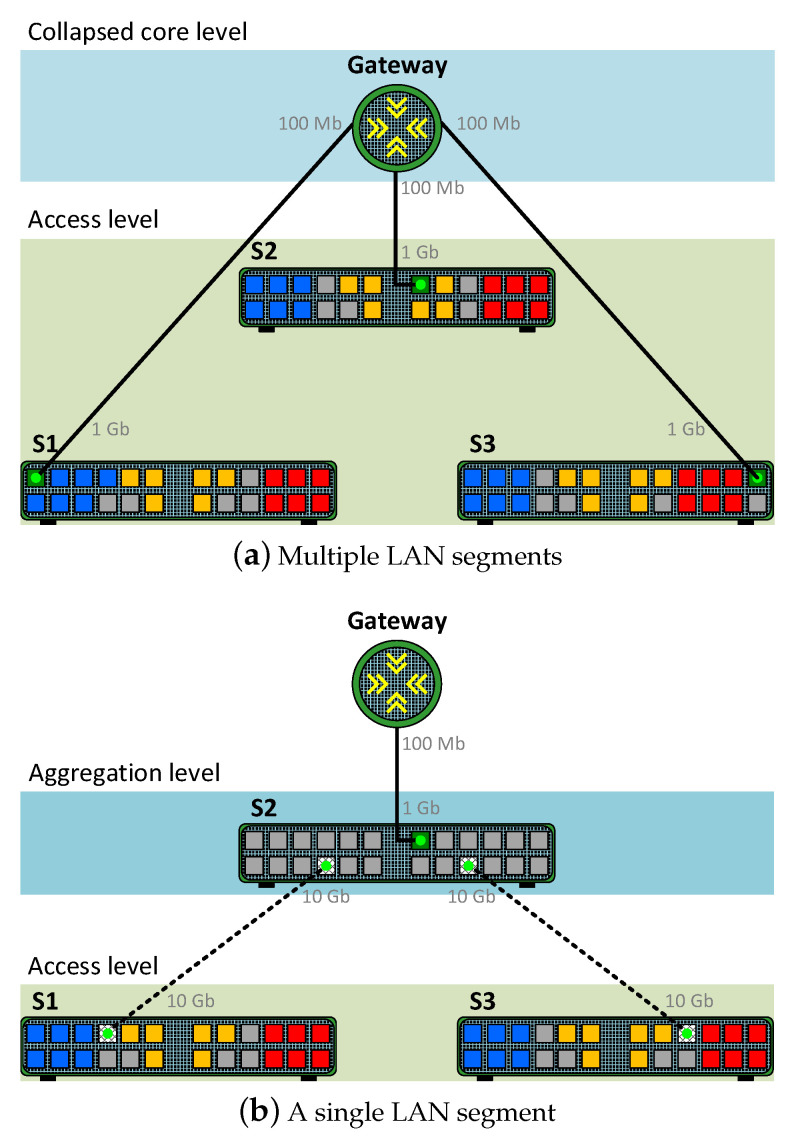
Scenarios of the emulated corporate network with PVLAN segmentation. Switch port key: community 1 (blue), community 2 (yellow), isolated (red), promiscuous (green), and trunks (dotted).

**Figure 6 sensors-23-01747-f006:**
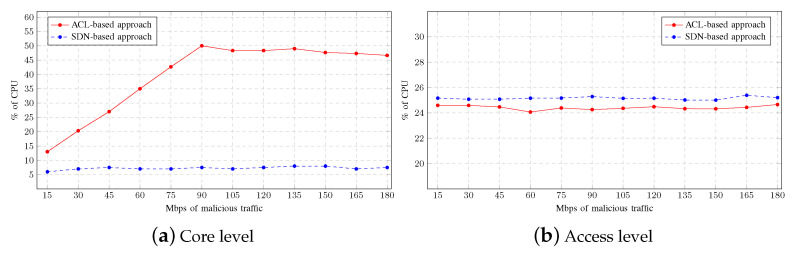
CPU usage on Multiple-LAN scenario.

**Figure 7 sensors-23-01747-f007:**
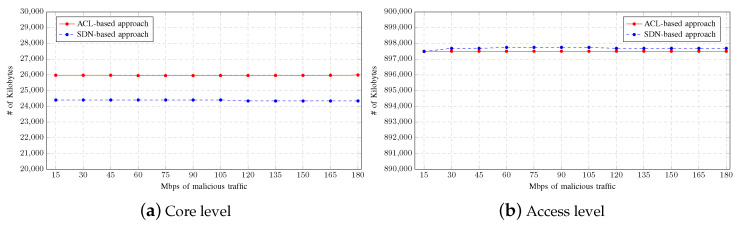
Memory consumption on Multiple-LAN scenario.

**Figure 8 sensors-23-01747-f008:**
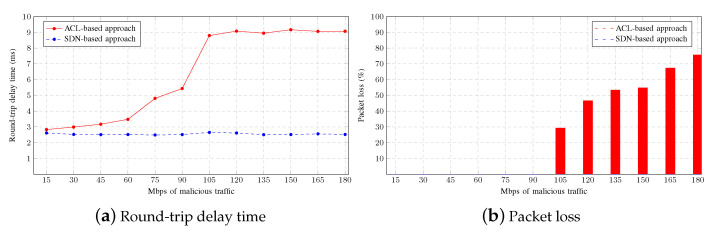
Legitimate user statistics.

**Figure 9 sensors-23-01747-f009:**
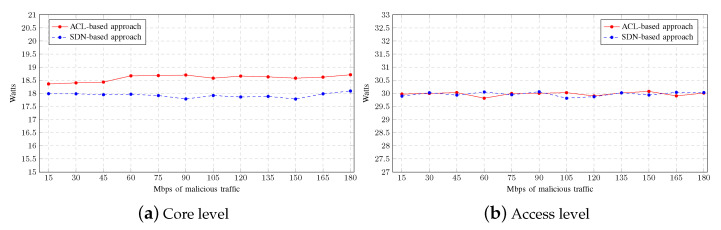
Power consumption on Multiple-LAN scenario.

**Figure 10 sensors-23-01747-f010:**
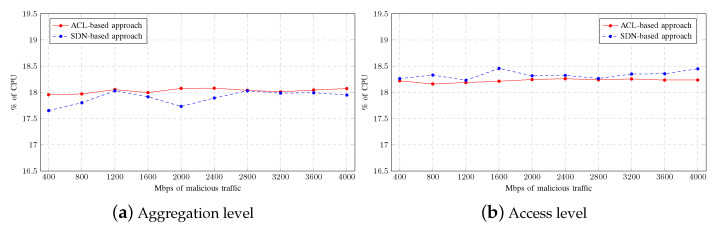
CPU usage on Single-LAN scenario.

**Figure 11 sensors-23-01747-f011:**
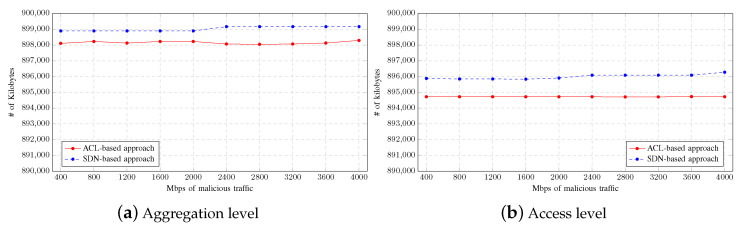
Memory consumption on Single-LAN scenario.

**Figure 12 sensors-23-01747-f012:**
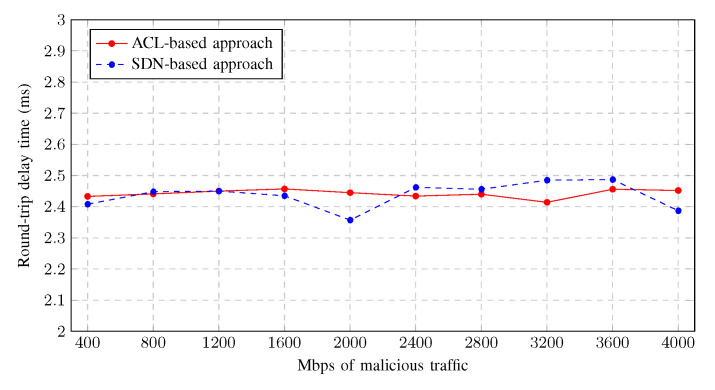
Legitimate user round-trip delay time.

**Figure 13 sensors-23-01747-f013:**
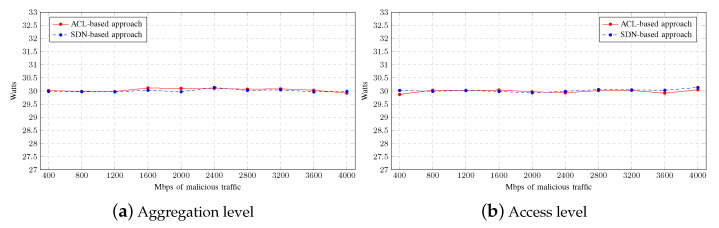
Power consumption on Single-LAN scenario.

**Figure 14 sensors-23-01747-f014:**
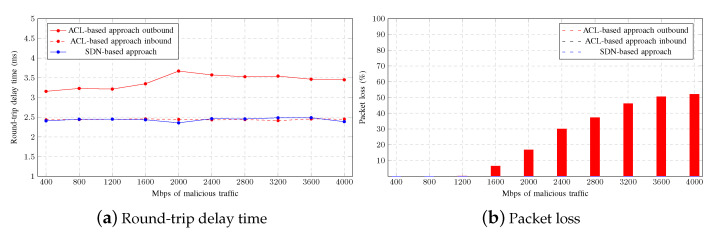
SDNs, inbound ACLs, and outbound ACL comparison.

**Table 1 sensors-23-01747-t001:** Experimentation settings.

Setting	Multiple-LAN Scenario	Single-LAN Scenario
Test duration	300 s	300 s
Number of repetitions	3	3
Device with ACL configured	Router	Switch (*S2*)
Device with Flow Table configured	Switches (*S1*, *S2*, *S3*)	Switches (*S1*, *S3*)
Background multicast traffic	8 Gbps	8 Gbps
Background legitimate traffic	1 Mbps	10 Mbps
Attackers located on each access switch	1	2
Malicious traffic increment	5 Mbps per attacker	100 Mbps per attacker

## Data Availability

The data used to support the findings of this study are available from the corresponding author upon request.

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
