# Peer review of "Performance Analysis of Software-Defined Networks to Mitigate Private VLAN Attacks"

_sensors, 2023, doi:10.3390/s23041747_

Round 1
Reviewer 1 Report
In this paper the authors compare security strategies to defend private VLANs using two mechanisms: a traditional ACL-based approach and a SDN configuration. The paper included emulation/test-bed studies and the attacks studied were associated with internal attacks over congested network configurations. The authors provided a detailed explanation of their proposal, and a description of the about performance advantages of both approaches.
I have several recommendations to improve the paper: in general, the standard of writing is high, but a final proof-editing is recommended. I did notice a few areas where the writing can be improved. Another recommendation is related to the approach used for comparison purposes: ACLs are rarely used on their own; even a traditional (or legacy) approach will include some Intrusion Detection System or firewall rules to complement ACL-based security. I know the study has already been carried out and it will be difficult to add a more realistic comparison but perhaps the authors can add something about these limitations of their work in the “Conclusions” section. There is another comment I’d like the authors to consider so the paper can be improved: the "Conclusions" section (section 6) should provide additional details about the benefits of the contribution; what are the implications for practitioners? I also recommend adding more detail to their brief discussion about future research that could advance the work done so far.
Reviewer 2 Report
This paper analyzes conventional and SDN-based mitigation strategies for PVLAN attacks. The paper is relatively well organized, so easy-to-follow. I suggest several questions and comments to improve the paper's reachability.
- For mitigation strategies, I believe the way of crafting ACL rules and OpenFlow rules would differ per design scheme. For example, is the one shown in this paper as the traditional strategy the only way to handle the attacks? Also, for the SDN-based strategy, one might match packets by L2 addresses only, but one might match up to L3 addresses. Also, the SDN controller not only controls switches but can control routers. Why is their control restricted on switches? Also, I'm curious why and how the two strategies compared in this study are designed and chosen.
- I recommend that authors summarize existing mitigation strategies in PVLAN attacks, which is not strongly focused on in Section 3.
- Regarding the evaluation, I'm curious about whether the rules of the SDN controller are installed proactively or reactively. If the rules were installed proactively, the communication between the controller and switches would not happen during packet flows, so the CPU or memory changes could be small.
- In Fig. 8, the difference in round trip time delay comes from switches' higher bandwidth and resources than routers. In fact, the difference between conventional and SDN-based schemes would be the existence of the SDN controller that manages rules. But the reason for the performance difference is from the device resources. I am not sure whether this is reasonable.
- In paragraph at # 47, page 2, the authors introduce the advantages of SDNs to motivate security consideration in SDN systems. As the beauty and applications of SDNs are much broader, I suggest that authors refer to the following studies for flexible and wide SDN usages (virtualization, time-sensitive networking, VANET, and AI systems).
n "A Case for SDN-based Network Virtualization." 2021 29th International Symposium on Modeling, Analysis, and Simulation of Computer and Telecommunication Systems (MASCOTS). IEEE, 2021
n "SDN-based configuration solution for IEEE 802.1 time sensitive networking (TSN)." ACM SIGBED Review 16.1 (2019): 27-32
n "Proactive congestion avoidance for distributed deep learning." Sensors 21.1 (2020): 174.
n Is SDN the solution for NDN-VANETs?." 2017 16th annual mediterranean ad hoc networking workshop (Med-Hoc-Net). IEEE, 2017
Round 2
Reviewer 2 Report
Thanks to the authors’ efforts in comprehending the comments in the previous review round, most concerns have been solved.
However, I am still not sure it is reasonable to compare overheads between two resources of different computing capacity.
